# Identifying Public Healthcare Priorities in Virtual Care for Older Adults: A Participatory Research Study

**DOI:** 10.3390/ijerph20054015

**Published:** 2023-02-23

**Authors:** Dai Pu, Victoria Palmer, Louise Greenstock, Cathie Pigott, Anna Peeters, Lena Sanci, Michele Callisaya, Colette Browning, Wendy Chapman, Terry Haines

**Affiliations:** 1School of Primary and Allied Health Care, Faculty of Medicine, Nursing and Health Sciences, Monash University, Frankston, VIC 3199, Australia; 2Monash Partners Academic Health Science Centre, Clayton, VIC 3168, Australia; 3Department of General Practice, Melbourne Medical School, Faculty of Medicine, Dentistry and Health Sciences, University of Melbourne, Melbourne, VIC 3010, Australia; 4Western Alliance Academic Health Science Centre, Warrnambool, VIC 3280, Australia; 5Institute for Health Transformation, Deakin University, Melbourne, VIC 3125, Australia; 6Peninsula Clinical School, Central Clinical School, Monash University, Frankston, VIC 3199, Australia; 7Menzies Institute for Medical Research, College of Health and Medicine, University of Tasmania, Hobart, TAS 7000, Australia; 8Institute of Health and Wellbeing, Federation University, Ballarat, VIC 3350, Australia; 9Centre for Research on Ageing, Health and Wellbeing, Australian National University, Canberra, ACT 2601, Australia; 10Centre for Digital Transformation of Health, University of Melbourne, Melbourne, VIC 3010, Australia

**Keywords:** telemedicine, COVID-19, public health, healthcare quality assurance, community-based participatory research, stakeholder engagement, aged, scaling up, shared decision making

## Abstract

There has been increasing adoption and implementation of virtual healthcare in recent years, especially with COVID-19 impacting the world. As a result, virtual care initiatives may not undergo stringent quality control processes to ensure that they are appropriate to their context and meet sector needs. The two objectives of this study were to identify virtual care initiatives for older adults currently in use in Victoria and virtual care challenges that could be prioritised for further investigation and scale-up and to understand why certain virtual care initiatives and challenges are prioritised over others for investigation and scale-up. Methods: This project used an Emerging Design approach. A survey of public health services in the state of Victoria in Australia was first carried out, followed by the co-production of research and healthcare priorities with key stakeholders in the areas of primary care, hospital care, consumer representation, research, and government. The survey was used to gather existing virtual care initiatives for older adults and any associated challenges. Co-production processes consisted of individual ratings of initiatives and group-based discussions to identify priority virtual care initiatives and challenges to be addressed for future scale-up. Stakeholders nominated their top three virtual initiatives following discussions. Results: Telehealth was nominated as the highest priority initiative type for scaling up, with virtual emergency department models of care nominated as the highest priority within this category. Remote monitoring was voted as a top priority for further investigations. The top virtual care challenge was data sharing across services and settings, and the user-friendliness of virtual care platforms was nominated as the top priority for further investigation. Conclusions: Stakeholders prioritised public health virtual care initiatives that are easy to adopt and address needs that are perceived to be more immediate (acute more so than chronic care). Virtual care initiatives that incorporate more technology and integrated elements are valued, but more information is needed to inform their potential scale-up.

## 1. Introduction

Virtual healthcare is rapidly becoming a vital part of how healthcare is delivered and received. The COVID-19 pandemic provided enormous impetus for healthcare services across all sectors to transition to virtual care alternatives in a short amount of time [1,2,3]. However, this bypasses the usually stringent quality control processes under which novel practices are adopted and/or scaled up in an organisation. Ideally, virtual care initiatives should be carefully planned and their barriers and facilitators assessed before they are implemented or scaled up, both to ensure that quality care is delivered via virtual platforms and that the initiatives maintain continued relevance in a future where virtual care becomes increasingly mainstream. Metropolitan Melbourne experienced the highest number of cumulative days in lockdown in the state of Victoria and Australia, which made the use of virtual care all the more relevant, as Victorians had to rely on technology where in-person visits were not possible. However, the rapid spread of infections and announcement of lockdowns prevented virtual care from undergoing the processes of evaluation and quality control traditionally practised prior to implementation and scale-up, outside of times of health system pressure.

One approach to ensure virtual care meets the needs and priorities of public health is the involvement of the target end-users from the community in participatory research. Co-design, co-production, and co-creation are terms that may be used interchangeably in participatory research [4,5,6] to describe contribution from stakeholder groups beyond traditional researchers, usually people who are affected by the issue under investigation, such as healthcare workers and patients and their care partners. These contributions need to be part of the core processes in the research or decision-making process that go beyond the roles of “participant” or “consultant”. Community engagement in health research synergistically combines the diverse experiences of those involved to improve the relevance and adoption of research [7,8], as well as stakeholder trust in research [7].

Engaging key stakeholders would be an appropriate method to understand and evaluate virtual healthcare initiatives from the perspectives of those who are the target end-users. First, because participatory research in healthcare holds a plethora of benefits, as already stated; second, because the “digital divide” that is intrinsic to challenges in virtual care [9] would be best represented by engaging those who currently, or will, experience and manage the divide. Stakeholders are inclusive of, but not exclusive to, consumers/patients, their families, and carers, healthcare staff who will provide care via virtual platforms, and administrative staff who will coordinate implementation of virtual care initiatives.

The digital divide can manifest as restrictions in perceptions and attitudes towards technology, the availability of equipment and infrastructure, the skills needed to use the technology, and opportunities to use the technology [10]. Virtual care initiatives that were newly implemented or expanded in response to the COVID-19 pandemic are highly susceptible to the challenge of restricted access due to existing digital divides [11,12]. Data on telemedicine access suggest that demographic variables are associated with different rates of usage [13,14,15,16,17], with older and culturally and linguistically diverse (CALD) individuals exhibiting reduced usage [13,16,17], which can compound existing inequalities in public healthcare. Therefore, there is a need to understand virtual care initiatives from multiple angles in order to identify successes, challenges, and potentially exemplary models of care that could be further developed for a post-pandemic world.

This study was part of a Victorian Department of Health-funded effort to aid in the recovery of healthcare following the extreme circumstances and pressure created by the COVID-19 pandemic. The primary aim of this project was to identify virtual care initiatives for older adults and virtual care challenges that could be prioritised for further investigation and scale-up through co-production. A secondary aim was to understand why participating stakeholders would prioritise certain virtual care initiatives and challenges over others during the co-production process.

## 2. Materials and Methods

### 2.1. Design

This project used an emerging design approach. Over a period of 4 months in 2021, research activities were carried out across three parts of the project. First, a survey was carried out to probe the breadth of virtual care initiatives in the state of Victoria in Australia. This was followed by 2 phases of priority-setting activities guided by the participatory health research paradigm [18], which will be referred to as co-production in this project to describe the processes in which virtual care initiatives for scaling up and further research were “produced” together with a group of healthcare providers and consumers. The descriptive survey and the co-production processes’ respective procedures and findings will be reported in a sequential manner to demonstrate how each step led to the next. Figure 1 visualises the flow of activities for this project.

### 2.2. Settings and Participants

This project was conducted in the State of Victoria in Australia, which is the second most populated state in the country. Health services in Victoria can consist of any combination of one to multiple tertiary hospitals and specialist and community health clinics [19]. The initial descriptive survey was conducted with staff from Victorian public health services. The 2-phase priority setting activities involved co-production with a group of stakeholders with a range of experience and knowledge of the Victorian healthcare system, including healthcare practitioners, healthcare consumers, carers of healthcare consumers, researchers, and health service managers. These stakeholders were recruited via the research team’s professional network, including consumers who were frequent contributors to the research and respondents who completed the survey and expressed further interest in the project. Invitations to be a part of the co-production team were sent to potential members whom the project team viewed as representatives of different healthcare disciplines and service settings. The invitation included the background of the project, anticipated time commitment as a part of the team, and remunerations for time. Stakeholders who responded affirmatively to the invitation were included in a mailing list for future contact. This group will be referred to as the co-production team from here on. The co-production team was contacted for each phase of co-production, and those who expressed interest in the described activity were provided with details of how to participate.

## 3. Descriptive Survey

### 3.1. Procedure

A survey was designed on the online survey platform Qualtrics to ask responders to nominate and describe individual virtual care initiatives at their place of work, including the initiatives’ objectives, breadth, maturity, success, barriers and enablers, and funding (Appendix A). The survey was disseminated via the Victorian State Department of Health to a news bulletin only accessible to the executives of Victorian public health services and the executives’ email inboxes for convenience of forwarding, with the instruction to further disseminate to staff with knowledge and experience of virtual care initiatives who could complete the survey. No further reminders or prompts were used to elicit responses. Snowball sampling was used by asking survey responders to nominate external personnel or organisations that were involved with the same or different virtual care initiatives, who were then sent the same survey.

### 3.2. Data Analysis

Virtual care initiatives were examined for distinct categories based on their intended function and target users. Thematic analysis was conducted through group discussions/interviews. Inductive analysis was used to identify discussion content that could be identified as themes within the experiences and perceptions of the implementation and potential scaling-up of different virtual care initiatives.

### 3.3. Results

A total of 41 survey responses were returned after the initial dissemination to health service executives, of which 38 responses were completed with analysable data. Snowball sampling returned an additional five survey responses, of which four were completed with analysable data. In total, 27 health services in Victoria were represented in the surveys. Figure 2 and Figure 3 show the geographical distribution of the responding health services and the position of the staff who completed the survey, respectively. Of the responders, one was aged 18–29 years (2.2%), four were aged 30–39 years (8.7%), fifteen were in each of the age ranges of 40–49 and 50–59 (32.6%), and seven were aged 60–69 (15.2%).

A total of 38 discrete virtual care initiatives were reported. Twenty-two of the virtual care initiatives reported in the survey operated in the specialist and/or primary care settings (58%), ten operated in the aged care setting (26%), and eight operated in the “Hospitals in the Home” setting (21%), which provides acute care outside of the hospital [20] (Figure 4). Two initiatives were explicitly reported to operate in more than one setting, but the description of multiple initiatives indicated that they were likely used across multiple settings. Figure 3 shows the target settings of the reported virtual care initiatives.

**Figure 2 ijerph-20-04015-f002:**
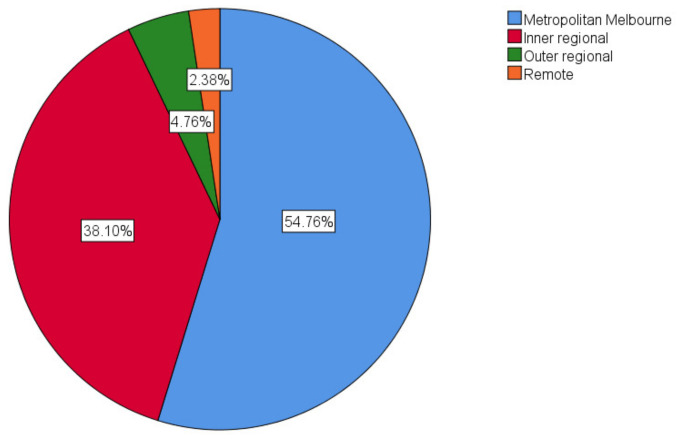
Location of the health service at which the survey responder is based classified according to the Australian Government’s workforce classification system [21].

**Figure 3 ijerph-20-04015-f003:**
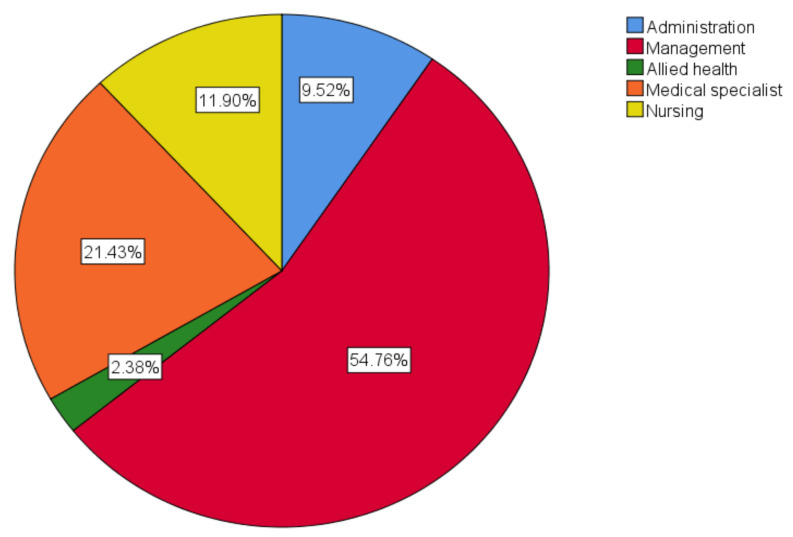
The primary role of the survey responder at the health service.

**Figure 4 ijerph-20-04015-f004:**
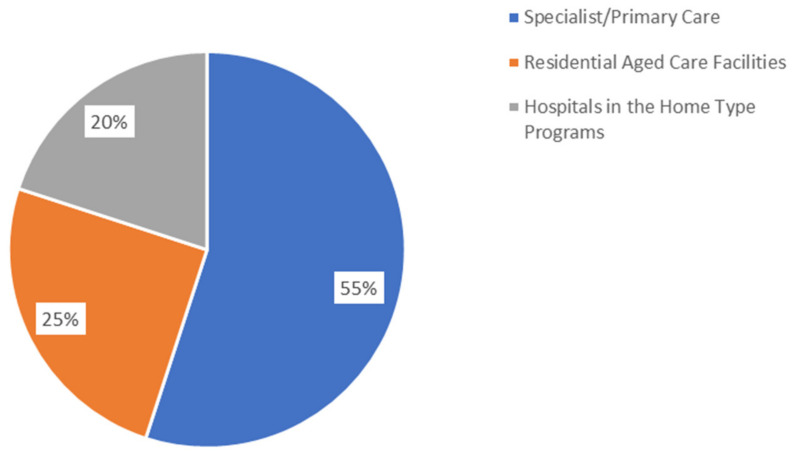
Settings in which virtual care initiatives were reported (Note: specialist and primary care were categorised together to reflect community health).

The virtual care initiatives were grouped into six distinct categories, with the novel individual initiatives within each category described below. A novel virtual care initiative was defined as one that made use of technology to deliver healthcare in an innovative manner or in a service setting that did not have an established use of virtual care. For this reason, telehealth initiatives that replaced in-person visits with any form of virtual communication without any other elements of innovation are not described here.

1.COVID-19-positive pathways and remote monitoring

COVID-19-positive pathways were implemented in multiple catchment areas in Victoria. COVID-19-positive patients were managed and monitored via telehealth and remote monitoring devices. However, these virtual care elements were part of the overall pathway. Five health services reported COVID-19-positive programs involving virtual care. 

2.Integrated Virtual Care

These initiatives involve multiple components of virtual care, including telehealth, remote monitoring and data collection, and data-driven feedback and decision-making. 

 Barwon Health’s personalised telehealth: One inner regional health service reported an integrated virtual care initiative with the aim of improving chronic disease management [22]. This initiative combined remote monitoring with telehealth.

3.Telehealth between one service provider and one consumer

These initiatives allow for communication between the consumer and the service provider when face-to-face visits are not possible; most virtual care initiatives reported in the survey belonged to this category, spanning across all regions and service settings in Victoria.

 My Emergency Doctor: A virtual emergency triaging service from the private sector was reported to be in use by one metropolitan health service. Further probing into this service revealed that it was used by multiple metropolitan health services in Victoria.

4.Telehealth amongst multiple service providers and one consumer

These initiatives allow for communication between multiple service providers and the consumer; service providers may be healthcare professionals such as general practitioners or specialists or support services such as interpreters for CALD consumers accessing care.

 The Northern Hospital’s virtual emergency department: One metropolitan hospital reported using an in-house virtual triaging system that facilitated patient care in the local catchment area. This allowed patients to access the service independently or via their general practitioner.Geri-connect: One inner regional health service reported a program to facilitate remote consultations with geriatricians for older adults and their primary carers in residential aged care facilities in regional Victoria.

5.Automated remote monitoring

These initiatives involve the use of a remote monitoring device that automatically collects biometric data via a wearable device and transfers the data to a service provider to inform medical decision making (e.g., a smartwatch that monitors heart rate throughout the day and sends the data via Bluetooth or an internet connection to a database).

 Murray Primary Health Network’s (PHN) chronic disease remote monitoring program: One inner regional health service reported being a part of this remote monitoring program for chronic diseases.Geri-connect 2: This was an extension of the Geri-connect telehealth initiative, where wearable devices were used for older adults in residential aged care facilities.myCare Companion Pandemic: One inner regional health service reported a program utilising both automatic and manual entry of data to monitor COVID-19 cases. This was independent of established COVID-19 care pathways in Victoria.

6.Manual remote monitoring

These initiatives involve the use of a remote monitoring device that requires the user to manually collect biometric data and upload the data to a database (e.g., the consumer uses a sphygmomanometer to measure blood pressure and enters the measurements into a database).

 myCare Companion Pandemic: This initiative included both automatic and manual entry of biometric data.

Eleven discrete challenges were identified in relation to the virtual care initiatives reported, and a number of these acknowledged specific challenges for older adult consumers. These challenges were:Consumer access to technology
Sensory impairments experienced by older adults may affect the successful implementation of telehealth;Residential aged care facilities may experience resource and/or skill shortages in the form of technology and staff to facilitate virtual care use.Consumer knowledge and attitude
Older adults may require initial demonstrations of how to use the technologies employed for virtual care.Culturally and linguistically diverse (CALD) populations’ use of virtual careData sharing across services and settingsLack of funding for virtual careLack of infrastructure for virtual careSecurity and privacy while using virtual careLack of training for staff for virtual careTelehealth can be insufficient for patient assessmentDifficulty tracking virtual care workflowLow user-friendliness of virtual care platforms
Sensory impairments experienced by older adults may affect their access to the platforms

## 4. Co-Production of Priorities—Phase 1: Rating Surveys

### 4.1. Procedure

The categories of virtual care initiatives, challenges, and services gaps identified in the descriptive survey were presented to the co-production team via word documents. Documents for initiatives varied in length from 4 to 7 pages, as initiatives such as telehealth were more numerous and had more details to report compared to initiatives involving newer technologies such as remote monitoring. Challenges and services gap descriptions were limited to 1 page each. The templates used to report on each one are provided in Appendix B.

For each initiative category, the co-production team members were asked:(1)“How beneficial do you think it would be if this virtual care initiative were to be scaled up across Victoria?”(2)“How important is it that more information is gathered about this initiative before the Department of Health seeks to scale up across Victoria?”

For each challenge, the co-production team members were asked: (1)“How beneficial do you think it would be if this virtual care challenge were to be addressed across Victoria?”(2)“How important is it that more information is gathered about this challenge before the Department of Health seeks to address it across Victoria?”

For each question, co-production members were asked to give their response on an 11-point Likert scale from 0 to 10, 0 being least beneficial/important, 10 being most beneficial/important. The ratings were conducted online via the platform Qualtrics.

### 4.2. Data Analysis

The average rating out of 10 for each question was calculated across all responses. Each virtual care initiative category and challenge had two ratings: the first was the perceived priority for scaling up or addressing, and the second was the perceived priority for further information gathering.

### 4.3. Results

Twenty-three complete responses were collected. Table 1 shows the demographic characteristics of the responders.

Telehealth initiatives were rated with the highest priority for scaling up, with remote monitoring initiatives rated with the lowest priority (Table 2). In contrast, remote monitoring initiatives were rated with the highest priority for further information gathering (Table 3). The same three challenges were rated with top priority for both addressing and further information gathering; these were “data sharing across services and settings”, “user-friendliness of the platforms”, and “consumer access to technology” (Table 2 and Table 3).

## 5. Co-Production of Priorities—Phase 2: Workshop Meeting

### 5.1. Procedure

The co-production team was asked to attend a workshop meeting conducted over an online videoconferencing platform. Co-production members who agreed to attend were sent a three-page word document that summarised the eight novel individual virtual care initiatives identified in the first survey of this project (Appendix C). This was different from previous documents, which had provided information on virtual care initiative categories rather than individual initiatives and their features. This time, co-production members were presented with the priority ratings for each virtual care initiative category and challenge and asked to discuss their motivations for their ratings from phase 1. The online platform MURAL was used to provide a visual representation of the virtual care initiative categories and individual initiatives (Figure 5). This platform allowed attendees to interact with the visual representations and provide written input in an anonymous manner.

At the end of the workshop, all attendees were asked to nominate the top three novel virtual care initiatives they would prioritise for research and potential scale-up. The workshop was video recorded.

### 5.2. Data Analysis

The workshop meeting was analysed in two ways: (1) thematic analysis of how priorities were identified and (2) tallying the number of votes for each novel virtual care initiative. Thematic analysis was conducted by the first author, who played back the workshop recording and referred to notes made during the workshop by three other project team members who were in attendance.

### 5.3. Results

Seventeen co-production team members attended the workshop, including three allied health clinicians, one general practitioner, two medical specialists, three consumers of healthcare, four health service managers, one health informatics researcher, two regional health service staff, and one nursing and midwifery researcher.

Three themes and three sub-themes were identified for how priorities were set:Theme 1: The breadth of reach of the initiative

High priority was assigned to virtual care initiatives that were perceived to have a wide breadth of reach. Telehealth initiative types were highly prioritised as they were considered in relation to what would be best for the population as a whole compared to what specific populations may most benefit from. Conversely, remote monitoring virtual initiatives were least prioritised for the same reason.

Sub-theme 1.1: Difficulty identifying the specificity of the initiative

While it was acknowledged that initiatives that utilise remote monitoring devices could be targeted at specific clinical populations, the amount of choice and products currently on the market made it difficult to pinpoint exactly how “what” could help “who”.

Theme 2: Addressing immediate/urgent healthcare needs

High priority was assigned to virtual care initiative types that were perceived to address immediate and/or urgent needs currently in public healthcare. Novel virtual care initiatives that addressed urgent care, e.g., emergency care, were more highly prioritised compared to remote monitoring and/or integrated programs that incorporated remote monitoring for chronic diseases, with the benefits of urgent care initiatives being more “visible”.

Theme 3: Practicalities of scaling-up the initiative

Higher priority was assigned to virtual care initiatives that were perceived to have lower costs; lower priority was assigned to initiatives that may entail additional effort from healthcare providers and consumers to learn new skills or acquire new equipment. 

Sub-theme 3.1: High costs

Higher costs associated with implementing and scaling up an initiative were of concern to the co-production members. This may be associated with the purchase of new equipment (e.g., remote monitoring devices), especially for rural and regional consumers. Direct comparisons were made between two similar virtual care initiatives that had the same purpose (virtual emergency triaging), with one in the private sector and one in the public sector. The additional costs perceived to be associated with the private model led to its lower prioritisation.

Sub-theme 3.2: Demand for digital and technological literacy

Virtual care initiatives with higher technical knowledge and skill demands on healthcare consumers and providers were given relatively lower priority. This was most relevant to remote monitoring, where it was raised that consumers may need a minimum level of digital and technological literacy to access certain smartphone applications, and providers needed to establish a clear workflow to process the remotely monitored data to inform clinical decisions.

At the end of the workshop meeting, all attendees were asked to nominate up to three novel virtual care initiatives that they would prioritise for scaling up out of the eight options. The eight options and the number of votes they received are listed below, in order of highest to lowest votes:Virtual emergency department (11 votes)My Emergency Doctor (9 votes)Barwon’s chronic disease management program (5 votes)Geri-Connect 1.0 (3 votes)Geri-Connect 2.0 (2 votes)Murray PHN’s remote monitoring of chronic conditions (2 votes)myCare Companion Pandemic (0 votes)COVID-pathways (0 votes)

## 6. Discussion

This study was part of an overall recovery effort in the state of Victoria in Australia in response to the COVID-19 pandemic. An emerging design approach was chosen to achieve this aim, given the unknowns in this arena. The major finding of this study was that stakeholders placed a higher priority on healthcare initiatives that address areas of perceived immediate and urgent need, in this case, virtual emergency care. The timing of this study coincides with media reports of overburdened emergency departments during the COVID-19 pandemic in Australia [23] may have contributed to this prioritisation. Nonetheless, this prioritisation demonstrates shared interest in virtual emergency care, which had been gaining momentum worldwide even prior to the COVID-19 pandemic. Virtual emergency care is a care model that has numerous and wide knowledge and evidence gaps, where challenges exist for both practitioners who will need to be educated for its use and consumers with complex social determinants of health that will impact their access to virtual care [24].

Another major finding was the identification of data sharing across services and settings as a priority challenge in virtual care. This challenge is one that affects the overall healthcare move to better-integrated care, which has been identified by the Australian Government’s Productivity Commission as one of the key challenges facing the healthcare system [25]. The Victorian government has highlighted the importance of achieving “greater integration of care” since 2015 [26]. Effective and secure data sharing across services and settings can allow the same patient receiving care from multiple sources to receive coordinated and timely care and reduce the amount of time and resources used to chase information and on redundant services (for example, saving time and money on tests that have already been conducted by another healthcare practitioner). There are technical, privacy, and security issues related to this challenge that will need to be addressed [27,28].

Virtual care has been suggested to be viewed through a socio-technical lens in order to fully understand its facilitators and barriers [29]. Four “blind spots” have been identified in their adoption: over-hyping of their positives, gaps in the evidence base, not addressing the impacts of social determinants of health, and lack of co-production with communities [29]. This study attempted to address the lack of co-production with the community in order to identify practice and research priorities, and in doing so, also identified social determinants of health relevant to the adoption and use of virtual care by providers and consumers. The virtual care challenges identified in this study reflect the combination of factors that influence how different stakeholders view virtual healthcare [30], but an implicit social determinant blind spot persisted: the lack of consideration of virtual care for older adults.

While this study began with the intention to focus on virtual care for older adults, the findings of the initial survey were reflective of virtual care initiatives in healthcare in general. This may be indicative of the shared underlying challenges for all virtual healthcare initiatives during their adoption and implementation, even when the initiatives are targeted at specific populations. However, it is likely that the low number of older adults in the initial survey responders (7 out of 38) and their position as service providers instead of consumers limited the perspective of how challenges may manifest for older adults using virtual care. The ideal virtual care should be broad in application and meet the shared needs of the population, a view shared by the co-production team in this study, but there are undeniable challenges specific to older adults and their use of virtual healthcare and technology overall [31,32]. To identify and address these challenges within the context of individual virtual care initiatives, older adults who are service providers and consumers should be part of the co-production process of generating solutions and novel designs.

### 6.1. Future Directions

Having identified specific priorities in virtual care initiatives and their relevant challenges, the next steps should evolve around addressing these priorities. Virtual care in emergency medicine is gaining traction internationally and in Australia, in particular, spurred on by the COVID-19 pandemic [33]. Given the relative novelty of this model of care, an evidence base should first be established with regard to its cost-effectiveness for patient safety, outcomes, and impacts on healthcare processes. The same can be said of remote monitoring technology in healthcare. The challenge of data sharing should also be focused on as a major challenge in virtual healthcare. Participatory research should be an integral part of all efforts to gather additional information to inform health service evaluation and scale-up.

### 6.2. Global Applicability

The methodology and findings of this study are relevant to public health research and priority setting for the global audience. We sought to identify existing practices and challenges in a mature public health setting and utilised an emerging design approach to use the most suitable methods to further investigate the initial findings. This allowed us to narrow the focus for future research and quality improvement programs to a small number of virtual care practices without “reinventing the wheel”. The processes were led by co-production with members of the scientific and healthcare community and broader society, with the project researchers acting as information “conductors”. These processes can be replicated in any other public health setting where existing practices and challenges need to be optimised and addressed, and a democratic approach is desired instead of allowing the process to be dictated by a few researchers and managers. While this study was conducted in a single state in Australia, the state (Victoria) has a mix of populous metropolitan settings and more remote health services in rural regions that are relatively siloed from each other, which were all represented in this study. The virtual care topic will also be highly relevant to public health researchers and practitioners around the world due to the global influence of technology advancement and COVID-19. The feasibility of this study demonstrates the potential for this co-production approach to be more widely utilised in countries that have similar public health landscapes.

### 6.3. Limitations

This study identified virtual care initiatives in public health services in the state of Victoria in Australia, which may not be representative of the private sector or other states and territories. The survey distribution approach in the first part of this project relied on health service CEOs to disseminate the survey to their staff, which meant that it was unknown how many and what type of staff received the survey in total and what proportion completed the survey. The 27 health services that reported virtual care initiatives in this project is 35.5% of the total 76 health services in Victoria, which limits the representativeness of the virtual care initiatives found for the entire state. The emerging design approach that was used allowed for flexibility of research targets and activities but also limited the depths to which each virtual care initiative could be explored, especially from the perspectives of older adults. While consumers were represented in the co-production process, overall, they were a smaller group compared to health practitioners. Consumers with culturally and linguistically diverse backgrounds were also not represented due to the research activities being carried out entirely in English. The virtual care initiatives and related challenges identified in this study should be investigated further to continue the effort of optimising virtual healthcare.

## 7. Conclusions

A large number of virtual care initiatives targeted at older adults have been implemented both prior to and as a response to the COVID-19 pandemic. Most of these aimed to facilitate communication and replace in-person visits in primary care and specialist care settings. Eight novel virtual care initiatives that made use of technology to deliver healthcare in an innovative manner or in a service setting that did not have an established use of virtual care were identified. Stakeholder consultation led to the identification of virtual emergency care as a priority initiative due to its potential to address healthcare needs that were perceived as urgent or immediate. Future research should focus on remote monitoring technology and how it can be used to improve health outcomes. Health services should utilise stakeholder co-production during decision-making processes to determine research and practice priorities.

## Figures and Tables

**Figure 1 ijerph-20-04015-f001:**
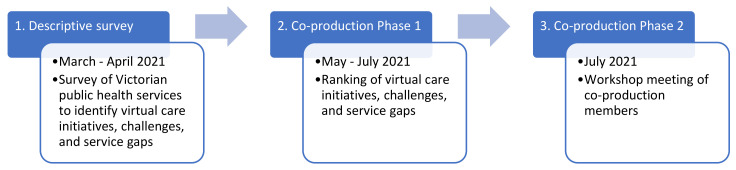
Research activity workflow.

**Figure 5 ijerph-20-04015-f005:**
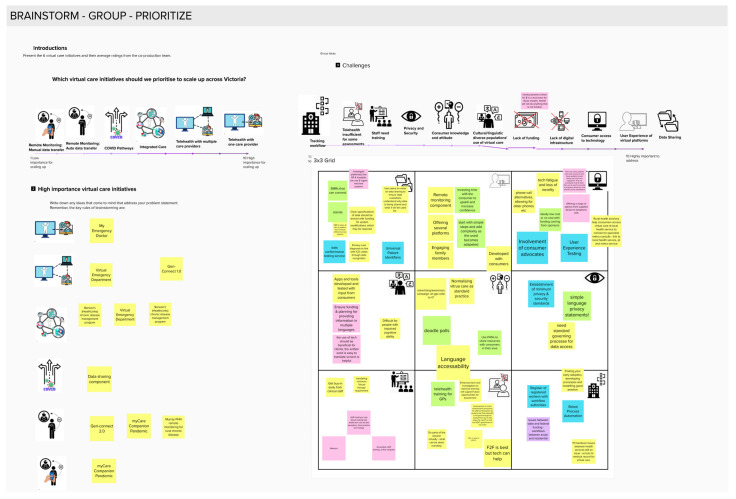
Online visual platform to support group co-production processes and anonymous input.

**Table 1 ijerph-20-04015-t001:** Demographics of the co-production team members who completed the ratings.

**Gender**	4 Males (17.4%)
	19 Females (82.6%)
**Age Groups**	5 aged 30–39 years old (21.7%)
	10 aged 40–49 years old (43.5%)
	4 aged 50–59 years old (17.4%)
	3 aged 60–69 years old (13%)
	1 aged 70 years or older (4.3%)
**Geographical Region**	20 Metropolitan Melbourne (87%)
	1 Inner Regional (4.3%)
	2 Unreported (8.7%)
**Professional Role**	4 Managers (17.4%)
	4 Allied health workers (17.4%)
	1 General Practitioner (4.3%)
	3 Medical Specialists (13%)
	1 Nurse (4.3%)
	9 Researchers (39.1%) *
	1 Consumer (4.3%) *
**Time spent in professional role**	4 spent less than 12 months (17.4%)
	7 spent 1–7 years (30.4%)
	5 spent 3–5 years (21.7%)
	3 spent 5–10 years (13%)
	4 spent more than 10 years (17.4%)

* Two responders who were recruited to the co-production team in the role of consumers self-identified as researchers during the survey completion.

**Table 2 ijerph-20-04015-t002:** Perceived priority of virtual care initiative categories and challenges for scaling up or addressing, listed in order of highest to lowest priority.

Virtual Care Initiative Categories (Average Rating)	Virtual Care Challenges (Average Rating)
Telehealth between one service provider and one consumer (8.7) Telehealth amongst multiple service providers and one consumer (8.6) Integrated care (8.1) COVID-19 Pathways (7.4) Remote monitoring with automated data transfer (7.3) Remote monitoring with manual data transfer (6.7)	Data sharing across services and settings (9.0) Low user-friendliness of the platforms (8.7) Consumer access to technology (8.5) Lack of infrastructure for virtual care (8.0) Lack of funding for virtual care (8.0) CALD * populations’ use of virtual care (8.0) Consumer knowledge and attitudes (7.9) Security and privacy while using virtual care (7.8) Lack of training for staff for virtual care (7.7) Telehealth can be insufficient for assessments (7.0) Difficulty tracking virtual workflow (5.4)

* CALD = culturally and linguistically diverse.

**Table 3 ijerph-20-04015-t003:** Perceived importance for further information gathering for virtual care initiative categories and challenges, listed in order of highest to lowest importance.

Virtual Care Initiative Categories (Average Rating)	Virtual Care Challenges (Average Rating)
Remote monitoring with automated data transfer (8.6) Integrated care (8.3) Remote monitoring with manual data transfer (7.9) Telehealth amongst multiple service providers and one consumer (7.2) Telehealth between one service provider and one consumer (7.0) COVID-19 Pathways (6.6)	Low user-friendliness of the platforms (8.3) Data sharing across services and settings (8.0) Consumer access to technology (7.9) Lack of funding for virtual care (7.7) Consumer knowledge and attitudes (7.6) CALD populations’ use of virtual care (7.3) Telehealth can be insufficient for assessments (7.2) Lack of training for staff for virtual care (7.2) Security and privacy while using virtual care (7.0) Lack of infrastructure for virtual care (6.5) Difficulty tracking virtual workflow (5.7)

## Data Availability

The data for this study are available upon reasonable request to the corresponding author.

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
