# Peer review of "Identifying Public Healthcare Priorities in Virtual Care for Older Adults: A Participatory Research Study"

_ijerph, 2023, doi:10.3390/ijerph20054015_

Round 1

Reviewer 1 Report

The paper is generally well written. There are some minor grammatical errors that should be addressed. The topic also addresses an important area, that given the impact of COVID and the rapid increase in virtual care initiatives, it is important to capture these programs. 

The introduction is generally good, painting the picture and the importance of understanding the VC priorities. At one point, it outlines the aim (line 65) of the study, and then does so again at line 99, I found this confusing, and don't think the study aim needs introducing at line 65, as it feels shoed in too early, when the authors are still outlining the current evidence and literature. 

I think there are other limitations (limitations section could be expanded) worth mentioning, there is no consent/participation rate due to the nature of survey distribution, so hard to know how impactful/relevant findings across across the 27 health districts. The methods could provide more detail, was the survey only sent out once, no reminders distributed? Also what strategies did they use if any to target the older population. Also, it used the term consumer views at times, but isn't it primarily a staff survey (sent to Health Executives) based on the methods, so limited consumer (patient) voice  (if this is not the case, please make this clearer with highlighting the specific strategies used to engage consumers?). Also was the survey/methods based on any other validated approaches (wasn't clear what evidence the survey built on? - the approach is clear (participatory),but not what the questions were based on etc.)

The number of responses feels like a low number (38) given the many health districts that were targeted, and it would be good to acknowledge this, or explain to the reader why they do not consider this to be an issue (for example, there are limited Virtual services/ initiatives, which may explain why only 38 responses across 27 health districts). 

The second phase of the methods with the consultation online exercise is also important to further explore the findings (challenges etc. and preferred programs for upscale), this could be made a bit clearer in the methods section. Also, acknowledging that the challenges and data reported is only based on 18 participatory views, although it is good that they had multiple disciplines. Maybe the study could suggest future in-depth interviews or additional focus groups to really explore the challenges in greater detail, as it feels like they have started to unearth the issues/areas but further research may help explore their findings?

I found the format of the paper a bit confusing, as the authors presented the methods of the survey and then results of one approach and then the methods of the second approach and results - rather than sectioning it off by Methods (all methods), Results (all results Part A and Part B), but I will leave that to the journal to decide if they would like formatting changes. 

I think the conclusion could be stronger and there are several grammatical errors in it, I would advise the authors to strengthen the conclusion and also focus it on the implications of this research for the health services and for virtual health, and next steps. 

Finally, the paper acknowledges this, but it doesn't really address its primary aim, with the research not that focussed on aged care/older patient population, and more a general exercise is identifying virtual programs in the health areas. 

I think the paper overall is valuable research, summarising virtual care initiatives and categorising these initiatives (and very relevant), but just some potential areas to acknowledge some potential limitations and some structural considerations in the manuscript. 

Author Response

The paper is generally well written. There are some minor grammatical errors that should be addressed. The topic also addresses an important area, that given the impact of COVID and the rapid increase in virtual care initiatives, it is important to capture these programs. 

  • Thank you for your feedback, time and effort in reviewing this manuscript. The manuscript has been checked for language issues during revision.

The introduction is generally good, painting the picture and the importance of understanding the VC priorities. At one point, it outlines the aim (line 65) of the study, and then does so again at line 99, I found this confusing, and don't think the study aim needs introducing at line 65, as it feels shoed in too early, when the authors are still outlining the current evidence and literature. 

  • This section (lines 65-69) has been deleted from the paragraph based on this suggestion to avoid repetitions and improve flow.

I think there are other limitations (limitations section could be expanded) worth mentioning, there is no consent/participation rate due to the nature of survey distribution, so hard to know how impactful/relevant findings across the 27 health districts.

  • Additional content has been added to the limitations section: “The survey distribution approach in the first part of this project relied on health service CEOs to disseminate the survey to their staff, which meant that it was unknown how many and what type of staff received the survey in total, and what proportion completed the survey. The 27 health services that reported virtual care initiatives in this project is 35.5% of the total 76 health services in Victoria, which limits the representativeness of the virtual care initiatives found for the entire state.”

The methods could provide more detail, was the survey only sent out once, no reminders distributed?

  • No reminders were given, details added to page 4 section 3.1 to explain this

Also what strategies did they use if any to target the older population.

  • The survey did not target older adults specifically, only health service staff who could report on virtual care initiatives that aimed to target older adults. This was explained in section 3.1.

Also, it used the term consumer views at times, but isn't it primarily a staff survey (sent to Health Executives) based on the methods, so limited consumer (patient) voice  (if this is not the case, please make this clearer with highlighting the specific strategies used to engage consumers?).

  • We did not seek to represent consumer views in the survey and reporting of the survey, these were in the co-production components of the study. Section 2.2 provides the distinction between the components.

Also was the survey/methods based on any other validated approaches (wasn't clear what evidence the survey built on? - the approach is clear (participatory), but not what the questions were based on etc.)

  • The survey questions were not based on any one framework, they were designed by the project team to gather the information we needed to address the project objectives

The number of responses feels like a low number (38) given the many health districts that were targeted, and it would be good to acknowledge this, or explain to the reader why they do not consider this to be an issue (for example, there are limited Virtual services/ initiatives, which may explain why only 38 responses across 27 health districts). 

  • This has been addressed by additions to the limitations section: “The survey distribution approach in the first part of this project relied on health service CEOs to disseminate the survey to their staff, which meant that it was unknown how many and what type of staff received the survey in total, and what proportion completed the survey. The 27 health services that reported virtual care initiatives in this project is 35.5% of the total 76 health services in Victoria, which limits the representativeness of the virtual care initiatives found for the entire state.”

The second phase of the methods with the consultation online exercise is also important to further explore the findings (challenges etc. and preferred programs for upscale), this could be made a bit clearer in the methods section. Also, acknowledging that the challenges and data reported is only based on 18 participatory views, although it is good that they had multiple disciplines. Maybe the study could suggest future in-depth interviews or additional focus groups to really explore the challenges in greater detail, as it feels like they have started to unearth the issues/areas but further research may help explore their findings?

  • The need for more in-depth investigations was acknowledged in section 6.1 and limitations. The additional info added to the limitations section detailed above also adds to this.

I found the format of the paper a bit confusing, as the authors presented the methods of the survey and then results of one approach and then the methods of the second approach and results - rather than sectioning it off by Methods (all methods), Results (all results Part A and Part B), but I will leave that to the journal to decide if they would like formatting changes. 

  • We wanted to reflect the emerging designs approach in how we reported the project, as we did not decide on the exact procedures of the co-production phases until after the survey was completed. As the methods had to be described for three parts of the project, we also felt that keeping the methods and their respective results together would help with readability. Nonetheless, thank you for your effort in tracking our format. We have also added clarifications in section 2.1 Design to let readers know about this structure before we begin to describe them.

I think the conclusion could be stronger and there are several grammatical errors in it, I would advise the authors to strengthen the conclusion and also focus it on the implications of this research for the health services and for virtual health, and next steps. 

  • This section has been edited for language.
  • End of conclusion reworded to be more specific and targeted at future research and health service actions: “A large number of virtual care initiatives targeted at older adults have been implemented both prior to and as a response to the COVID-19 pandemic. Most of these aimed to facilitate communication and replace in-person visits in the primary care and specialist care settings. Eight novel virtual care initiatives that made use of technology to deliver health care in an innovative manner, or in a service setting that did not have an established use of virtual care, were identified. Stakeholder consultation led to the identification of virtual emergency care as a priority initiative due to its potential to address health care needs that were perceived as urgent or immediate. Future research should focus on remote monitoring technology and how they can be used to improve health outcomes. Health services should utilise stakeholder co-production during decision-making processes to determine research and practice priorities.”

Finally, the paper acknowledges this, but it doesn't really address its primary aim, with the research not that focussed on aged care/older patient population, and more a general exercise is identifying virtual programs in the health areas. 

  • This was something that became apparent as the project was ongoing, but we continued to include and prompt for focus on older adults throughout all research activities. While there are documented challenges specific to older adults and their use of virtual care, we believe the results of this project are also indicative of an overall trend of technology acceptability and literacy, even amongst older adults.

Reviewer 2 Report

This paper reports on a co-production approach to identifying priority telehealth services in Victoria for upscaling and implementing. The question is very relevant and the overall approach is good. Some of the findings and prioritisation are useful. However, there are the following major issues:

1. The paper says it is focussing on older people's services, but the services seem fairly generic and there is little evidence the authors tried to recruit older people to understand their views (the lack of them is listed as a limitation, although I can only see 4 aged over 60 in the table not the 7 described). It would also have been relevant to have residential care staff's views, however I can not see any reference to them being approached or taking part. Recruitment of the stakeholder panel needs much more description - it is difficult to know exactly how these people were recruited and how many from where (e.g. why 1 consumer in the first survey and 3 in the workshop?).

2. The consumer voice is almost entirely absent, which undermines the idea of this being co-production. Whilst the authors have recruited a good range of healthcare professionals, co-production should ensure those who voices are less often heard are involved. Given that the initial service survey and professional networks are listed as the sources of recruitment it does not seem surprising that few (older) consumers were recruited. The authors need to justify why they did not specifically try and reach out to these populations e.g. through local community groups, newspapers etc. Similarly, we have no idea from the current manuscript how diverse the group was culturally and linguistically, despite the introduction and discussion highlighting that these are characteristics associated with the digital divide. This should be reported, and again, justified why no attempts were made to get a diverse sample. 

Minor points:

- There does not seem to have been any option for respondents to identify further potential challenges , only to rank existing ones

- L249-251 - clearly text left in from a template that needs deleting. 

- Virtual care initiatives outlined would be easier to read in a table

- Unclear why specialist and community care initiatives were grouped together - these would seem quite diverse (although I come from a UK perspective where there is a strong divide)

- L115-157 is confusingly worded

- in the first survey, how many people were originally emailed to prior to snowballing?

L69 ender users should be end users 

- analysis details need reporting. Inductive analysis (L141) says nothing about whether it was thematic or content. 

- was the workshop recorded or just detailed notes taken by facilitators? This is unclear. What exactly was analysed?

Author Response

  1. The paper says it is focussing on older people's services, but the services seem fairly generic and there is little evidence the authors tried to recruit older people to understand their views (the lack of them is listed as a limitation, although I can only see 4 aged over 60 in the table not the 7 described). It would also have been relevant to have residential care staff's views, however I can not see any reference to them being approached or taking part. Recruitment of the stakeholder panel needs much more description - it is difficult to know exactly how these people were recruited and how many from where (e.g. why 1 consumer in the first survey and 3 in the workshop?).
  • The virtual care initiatives nominated were not always focused on older adults, although we maintained and prompted for this focus throughout the research activities. As the reviewer stated, we tried to acknowledge this, but did not want to exclude any findings.
  • The 4 older adults were those in the co-production team who contributed to phase 1 of the co-production process (section 4, Table 1), the 7 older adults were the health service staff who nominated virtual care initiatives in the survey (section 3.3)
  • Residential care staff view would have been valuable, however this project was conducted from March to July 2021, and Melbourne was experiencing rising COVID infection rates and lockdowns that affected a high number of aged care facilities, resulting in low staffing. We did not approach aged care at the time with these considerations.
  • Details for stakeholder recruitment has been added to section 2.2 (page 3, lines 126-131): “These stakeholders were recruited via the research team’s professional network, including consumers who were frequent contributors to research, and from respondents who completed the survey and expressed further interest in the project. Invitations to be a part of the co-production team were sent to potential members that the project team viewed as representatives of different health care disciplines and service settings. The invitation included the background of the project, anticipated time commitment as a part of the team, and remunerations for time. stakeholders who responded affirmative to the invitation were included in a mailing list for future contact. This group will be referred to as the co-production team from here on. The co-production team was contacted for each phase of co-production, and those who expressed interest for the described activity were provided with details of how to participate.”

  1. The consumer voice is almost entirely absent, which undermines the idea of this being co-production. Whilst the authors have recruited a good range of healthcare professionals, co-production should ensure those who voices are less often heard are involved. Given that the initial service survey and professional networks are listed as the sources of recruitment it does not seem surprising that few (older) consumers were recruited. The authors need to justify why they did not specifically try and reach out to these populations e.g. through local community groups, newspapers etc. Similarly, we have no idea from the current manuscript how diverse the group was culturally and linguistically, despite the introduction and discussion highlighting that these are characteristics associated with the digital divide. This should be reported, and again, justified why no attempts were made to get a diverse sample. 
  • The research activities in this project were conducted in a relatively short time frame of ~ 4 months amid COVID lockdowns in the state of Victoria in 2021, the circumstances greatly limited the stakeholders we could access and our response rates. Consumers who were frequent contributors to research run by the project team were invited, but responses were limited and the consumer engagement panels we were planning to reach out to were not running during COVID. We understand that these explanations do not make up for the lack of consumer voice in this project, and have added edits in addition to those already in the manuscript to acknowledge the limitations: “While consumers were represented in the co-production process, overall they were a smaller group compared to health practitioners. Consumers with culturally and linguistically diverse backgrounds were also not represented due to the research activities being carried out entirely in English.” – page 12, section 6.3

- There does not seem to have been any option for respondents to identify further potential challenges , only to rank existing ones

  • The challenges came from health service staff who responded to the initial survey, and we wanted to limit the research to what was being used and experienced to inform scalability

- L249-251 - clearly text left in from a template that needs deleting. 

  • Thank you for spotting this, the paragraph has now been deleted.

- Virtual care initiatives outlined would be easier to read in a table

  • We tried to place this section in a table but the spacing did not look good, we’ve edited the font and formatting to improve readability

- Unclear why specialist and community care initiatives were grouped together - these would seem quite diverse (although I come from a UK perspective where there is a strong divide)

  • Primary and specialist care settings were grouped together to reflect community health services provided by health services/hospitals, while residential aged care and hospitals in the home were separated out to allow us to focus on older adults and home hospital care. This is briefly explained in the caption for figure 4.

 L115-157 is confusingly worded

  • Perhaps the reviewer meant lines 151-157? This section has been edited for easier reading, currently line 170-178, page 4.

- in the first survey, how many people were originally emailed to prior to snowballing?

  • We were unable to track this as the survey was circulated to health service CEOs who were asked to pass the survey to their staff. This is described in section 3.1 on page 4.

L69 ender users should be end users 

  • Thank you for spotting this, now fixed

- analysis details need reporting. Inductive analysis (L141) says nothing about whether it was thematic or content. 

  • Section 3.2 edited to: “Thematic analysis was conducted on group discussions/interviews. Inductive analysis was used to identify discussion content that could be identified as themes within the experiences and perceptions of the implementation and potential scaling-up of different virtual care initiatives.”

- was the workshop recorded or just detailed notes taken by facilitators? This is unclear. What exactly was analysed?

  • Analysis detailed added in section 5.2: “Thematic analysis was conducted by the first author, who played back the workshop recording and referred to notes made during the workshop by three other project team members who were in attendance.”

Reviewer 3 Report

The research work entitled: "Identifying public healthcare priorities in virtual care for older adults: a participatory research study", has as its first objective to identify virtual care initiatives for older adults and virtual care challenges that could be prioritized for further investigation and scale-up through co-production. A secondary aim was to understand why participating stakeholders would prioritize certain virtual care initiatives and challenges over others during the co-production process.

The topic addressed is of great interest and topicality and is very successful in suggesting the importance of the issues discussed to be taken into account by public health decision-makers.

The information presented is new in the international, post-COVID-19 environment and may be useful for better planning of virtual health care in future emergency situations.

Thus, the manuscript may be appropriate for the journal, although it should be considerably improved even though it is well organized especially in the methodology and results part.

The following are the main recommendations that I consider necessary to improve it.

The main limitation of the study is related to the fact that the information collected has been provided exclusively by health professionals. If an adequate representation of people from the community had been included, the work would provide greater evidence of the real possibilities that virtual care has for improving the care of people who cannot go to health centers. Since this limitation can no longer be solved, I recommend that it be expressed in the limitations section of the study. In a way, this limitation is exposed by the authors in recognizing the need to promote participatory research (line 429) as one of the future directions.

The introduction is clearly and briefly presented, justifying the need to know the resources available to carry out virtual care for the elderly.

The summary shows more clearly the separation between the methods and results sections.

In the Methods section, I recommend adjusting the sections by clearly separating the methods section from the results section. Results section starting on line 139 (not 3.2.-Data Analysis), but you may consider a major modification by more clearly separating the methods and results sections. It is necessary to highlight the sections following lines 174 and 225.

In conclusion, this is an interesting work for the readers of the journal, but I consider that it has to be improved in the aspects commented.

Author Response

The main limitation of the study is related to the fact that the information collected has been provided exclusively by health professionals. If an adequate representation of people from the community had been included, the work would provide greater evidence of the real possibilities that virtual care has for improving the care of people who cannot go to health centers. Since this limitation can no longer be solved, I recommend that it be expressed in the limitations section of the study. In a way, this limitation is exposed by the authors in recognizing the need to promote participatory research (line 429) as one of the future directions.

  • Although consumers are not highly represented, we do have consumer voices in the co-production components of this project. The period during which this study was conducted coincided with COVID spikes and lockdowns in Victoria, so recruitment overall was difficult. However, we acknowledge that this remains a limitation of the project. We have added to this acknowledge in the limitations section following feedback from all reviewers, the limitations section now reads: “This study identified virtual care initiatives in public health services in the state of Victoria in Australia, which may not be representative of the private sector or other states and territories. The survey distribution approach in the first part of this project relied on health service CEOs to disseminate the survey to their staff, which meant that it was unknown how many and what type of staff received the survey in total, and what proportion completed the survey. The 27 health services that reported virtual care initiatives in this project is 35.5% of the total 76 health services in Victoria, which limits the representativeness of the virtual care initiatives found for the entire state. The emerging design approach that was used allowed for flexibility of research targets and activities, but also limited the depths to which each virtual care initiative could be explored, especially from the perspectives of older adults. While consumers were represented in the co-production process, overall they were a smaller group compared to health practitioners. Consumers with culturally and linguistically diverse backgrounds were also not represented due to the research activities being carried out entirely in English. The virtual care initiatives and related challenges identified in this study should be investigated further to continue the effort of optimising virtual health care.”

The introduction is clearly and briefly presented, justifying the need to know the resources available to carry out virtual care for the elderly.

  • Thank you

The summary shows more clearly the separation between the methods and results sections.

In the Methods section, I recommend adjusting the sections by clearly separating the methods section from the results section. Results section starting on line 139 (not 3.2.-Data Analysis), but you may consider a major modification by more clearly separating the methods and results sections. It is necessary to highlight the sections following lines 174 and 225.

  • This feedback is noted, and is similar to feedback from another reviewers. We reported this project in its current format to reflect the emerging designs approach used, as we did not decide on the exact procedures of the co-production phases until after the survey was completed. Additionally, as the methods had to be described for three parts of the project, we also felt that keeping the methods and their respective results together would help with readability. We have also added clarifications in section 2.1 Design to let readers know about this structure before we begin to describe them.

Reviewer 4 Report

I have only minor comments:

How many participants did you plan to include in the study?

Have you considered including more people over 65 (older adults) in this study?

How stakeholders were recruited? (Line 124)

Author Response

How many participants did you plan to include in the study?

  • We did not have any sample size estimates for the initial survey, but we did want to have a minimum of 1-2 representatives of each type of health care discipline and setting and consumers in the co-production processes. This was achieved.

Have you considered including more people over 65 (older adults) in this study?

  • Yes, we did want to include more older people representatives in the co-production processes. However, this project was carried out during a period of ~4 months in 2021 when Victoria was severely affected by COVID infections and lockdowns, so consumers of all types were difficult to recruit, and the health professionals and researchers we did recruit tended to be younger than retirement age. This is a major limitation that has been pointed out by all reviewers and we have added the following to the limitations section: “While consumers were represented in the co-production process, overall they were a smaller group compared to health practitioners. Consumers with culturally and linguistically diverse backgrounds were also not represented due to the research activities being carried out entirely in English.”

How stakeholders were recruited? (Line 124)

  • This has been expanded on in section 2.2: “These stakeholders were recruited via the research team’s professional network, including consumers who were frequent contributors to research, and from respondents who completed the survey and expressed further interest in the project. Invitations to be a part of the co-production team were sent to potential members that the project team viewed as representatives of different health care disciplines and service settings. The invitation included the background of the project, anticipated time commitment as a part of the team, and remunerations for time. stakeholders who responded affirmative to the invitation were included in a mailing list for future contact. This group will be referred to as the co-production team from here on. The co-production team was contacted for each phase of co-production, and those who expressed interest for the described activity were provided with details of how to participate.”